# Whole Genome Analysis of Ovarian Granulosa Cell Tumors Reveals Tumor Heterogeneity and a High-Grade TP53-Specific Subgroup

**DOI:** 10.3390/cancers12051308

**Published:** 2020-05-21

**Authors:** Joline Roze, Glen Monroe, Joachim Kutzera, Jolijn Groeneweg, Ellen Stelloo, Sterre Paijens, Hans Nijman, Hannah van Meurs, Luc van Lonkhuijzen, Jurgen Piek, Christianne Lok, Geertruida Jonges, Petronella Witteveen, René Verheijen, Gijs van Haaften, Ronald Zweemer

**Affiliations:** 1Department of Gynaecological Oncology, UMC Utrecht Cancer Center, University Medical Center Utrecht, Utrecht University, 3584 CX Utrecht, The Netherlands; G.Monroe@umcutrecht.nl (G.M.); J.W.Groeneweg-11@umcutrecht.nl (J.G.); rene.h.m.verheijen@gmail.com (R.V.); R.Zweemer@umcutrecht.nl (R.Z.); 2Department of Genetics, Center for Molecular Medicine, University Medical Center Utrecht, Oncode Institute, Utrecht University, 3584 CX Utrecht, The Netherlands; J.Kutzera@umcutrecht.nl (J.K.); E.Stelloo@umcutrecht.nl (E.S.); G.vanHaaften@umcutrecht.nl (G.v.H.); 3Department of Obstetrics and Gynaecology, University Medical Center Groningen, University of Groningen, 9713 GZ Groningen, The Netherlands; s.t.paijens@umcg.nl (S.P.); h.w.nijman@umcg.nl (H.N.); 4Department of Gynecological Oncology, Centre for Gynaecological Oncology Amsterdam, Amsterdam University Medical Center, 1105 AZ Amsterdam, The Netherlands; h.s.vanmeurs@amsterdamumc.nl (H.v.M.); l.r.vanlonkhuijzen@amsterdamumc.nl (L.v.L.); 5Department of Obstetrics and Gynaecology, Catharina Hospital, 5623 EJ Eindhoven, The Netherlands; jurgen.piek@catharinaziekenhuis.nl; 6Department of Gynaecological Oncology, Centre for Gynaecological Oncology Amsterdam, The Netherlands Cancer Institute, Antoni van Leeuwenhoek Hospital, 1066 CX Amsterdam, The Netherlands; c.lok@nki.nl; 7Department of Pathology, University Medical Center Utrecht, Utrecht University, 3584 CX Utrecht, The Netherlands; G.N.Jonges@umcutrecht.nl; 8Department of Medical Oncology, University Medical Center Utrecht, Utrecht University, 3584 CX Utrecht, The Netherlands; P.O.Witteveen@umcutrecht.nl

**Keywords:** whole genome sequencing, granulosa cell tumors, ovarian cancer, FOXL2, tumor heterogeneity

## Abstract

Adult granulosa cell tumors (AGCTs) harbor a somatic *FOXL2* c.402C>G mutation in ~95% of cases and are mainly surgically removed due to limited systemic treatment effect. In this study, potentially targetable genomic alterations in AGCTs were investigated by whole genome sequencing on 46 tumor samples and matched normal DNA. Copy number variant (CNV) analysis confirmed gain of chromosome 12 and 14, and loss of 22. Pathogenic *TP53* mutations were identified in three patients with highest tumor mutational burden and mitotic activity, defining a high-grade AGCT subgroup. Within-patient tumor comparisons showed 29–80% unique somatic mutations per sample, suggesting tumor heterogeneity. A higher mutational burden was found in recurrent tumors, as compared to primary AGCTs. *FOXL2*-wildtype AGCTs harbored *DICER1*, *TERT*(C228T) and *TP53* mutations and similar CNV profiles as *FOXL2*-mutant tumors. Our study confirms that absence of the *FOXL2* c.402C>G mutation does not exclude AGCT diagnosis. The lack of overlapping variants in targetable cancer genes indicates the need for personalized treatment for AGCT patients.

## 1. Introduction

Adult granulosa cell tumors (AGCTs) belong to the sex cord-stromal tumors of the ovary and account for 2–5% of ovarian malignancies with an estimated incidence of 0.6–1.0 per 100,000 women per year worldwide [1,2,3,4]. Patients may develop symptoms, such as vaginal bleeding, caused by prolonged exposure to tumor-derived estrogen, which may result in early detection of the disease. However, AGCTs are usually not preoperatively suspected and a histopathological diagnosis is made after surgical resection of an ovarian mass. Granulosa cell tumors also exist as a juvenile subtype, which generally occurs in young women and represents only 5% of this tumor type [4]. AGCTs are microscopically defined by grooved, uniform and pale nuclei and a variety and mixture of histologic patterns can be found including microfollicular, trabecular, insular and diffuse patterns [4]. Although the disease is frequently described as indolent in behavior, recurrences occur in approximately 50% of the patients of whom 50–80% ultimately die of their disease [5,6,7]. Recurrences often require repeated debulking surgeries since alternative treatment options including chemotherapy, radiotherapy and hormone therapy have thus far shown only limited effect [8]. The current lack of effective systemic treatment emphasizes the need for novel therapeutic strategies. Molecular characterization of AGCTs could help to identify potential targets for treatment.

In contrast with other ovarian malignancies, AGCTs harbor a specific *FOXL2* c.402C>G point mutation (C134W), which has been reported in 94–97% of patients [6,9]. FOXL2 is a transcription factor involved in ovarian function and granulosa cell differentiation [10,11]. Until now, efforts to target this gene have not been successful. Previous studies on genomic alterations in AGCTs identified copy number gains in chromosome 12 and 14 and loss of chromosome 22 [12,13,14,15,16,17,18]. Furthermore, mutations were detected in genes that are known for their involvement in other cancer types, such as *TERT, KMT2D, PIK3CA, AKT1, CTNNB1* and *NR1D1* [13,14,19,20,21,22]. These studies included a subset of AGCTs within a larger ovarian cancer cohort and/or performed targeted or whole exome sequencing. Importantly, most studies did not analyze the corresponding normal reference DNA, essential for identifying true somatic variants.

We present the largest molecularly characterized cohort of AGCTs to date, in which we perform whole genome sequencing (WGS) on fresh frozen tumor material with matched normal reference DNA. We use this comprehensive method to investigate copy number changes, frequently mutated genes, mutational signatures and tumor heterogeneity. We define a subgroup of patients with high-grade AGCTs, harboring a pathogenic *TP53* mutation. In addition, we identify potential driver mutations in AGCTs without the *FOXL2* c.402C>G mutation, and find more variants in recurrent AGCTs as compared to primary tumors. Finally, we detect a high degree of intra-patient tumor heterogeneity.

## 2. Results

### 2.1. Description of WGS Cohort

WGS was performed on 46 fresh frozen tumor samples from 33 patients (Table 1). We analyzed 11 patients with a primary tumor and 22 patients with recurrent disease. Microscopically assessed average tumor purity was 80% (range 40–90%). The whole genome was sequenced to an average read depth of 35X per sample (range: 26–107X), with 97.5% of bases covered >10-fold (range: 93.9–98.3%, Appendix A). Matched normal reference DNA was obtained for all AGCTs. The median age at diagnosis was 53 years (range 29–75) and at the time of study, 21 patients had no evidence of disease, 11 were alive with disease and one patient had died of her disease. From five patients, tumors at multiple time points and/or multiple tumor locations were analyzed.

In our cohort, we detected a median number of somatic single nucleotide variants (SNVs) and small insertions/deletions (INDELs) of 3579 variants per genome (range 1346–21,452), of which 29 (range 13–238) were exonic, resulting in a tumor mutational burden (TMB) of approximately 1.25 per megabase (Mb, range 0.47–7.5). We identified 20 structural variants (SVs; excluding copy number variants (CNVs)) per sample (median, range 0–314). This low number of SVs indicates that DNA repair mechanisms are still intact in AGCTs. The median number of mutations (SNVs and INDELs) detected in primary tumors was 1.5× higher than reported in a previous WGS study including 10 primary AGCTs which identified 1578 variants per tumor (range 630–2706) [19]. The TMB in our AGCT cohort was comparable to the TMB reported in a previous whole exome sequencing based study (1.2 mutations per Mb in primary AGCTs and 2.1 mutations per Mb in recurrences) [13]. The number of variants detected in AGCTs falls within the same range of variants described in low grade serous ovarian cancer (median: 3064, range 1641–7398) [23].

Previous studies reported conflicting results on the difference in mutational burden between primary and recurrent tumors [13,14]. In our study, primary tumors harbored 2199 SNVs (median, range 1346–4120) and recurrent tumors 4279 SNVs per sample (median, range 2114–21,452; *p* < 0.001). In addition, primary tumors carried fewer SVs (median 10, range 2–34) as compared to recurrent tumors (median 22, range 0–314; *p* = 0.018). It is known that platinum-based chemotherapy can increase the number of variants [24]. However, when we compared the mutations in primary tumors with recurrences that had not been treated with chemotherapy, the number of variants was still significantly different (median 2199 SNVs versus 3634 SNVs, *t*-test *p*-value = 0.0009). These results suggest that recurrences, also when stratified for prior chemotherapy treatment, harbor significantly more variants than primary tumors. 

### 2.2. Common Copy Number Alterations in Chromosome 12, 14 and 22

Copy number analysis was performed on WGS data of 27 patients fulfilling the CNV caller pipeline requirements (see Methods). The majority of copy number alterations were duplications or losses of entire chromosome arms or chromosomes. In most patients, copy number loss of chromosome 22(q) (15/27, 56%) or gain of chromosome 14 (15/27, 56%) was found (Figure 1). Concurrent gain of chromosome 14 and loss of 22 was seen in 11/27 patients (41%). Loss of chromosome 16(q) was seen in 4/27 patients (15%) and gain of chromosome 1, 7, 8, 9, 12, 13, 14, 15, 18 or 20 were identified in at least 15% of patients. Concurrent copy number gains of chromosomes 8, 9 and 12 were detected in 7/27 patients (26%) and gain of both chromosome 18 and 20 was seen in 5/27 patients (19%). Within patients, copy number profiles remained stable between different time points and tumor locations (patient 8 and 13, Figure 1), or differed slightly between time points (patient 11 T1 versus T2 and T3). CNVs were detected in 7/9 (78%) patients with a primary tumor and in 16/18 (89%) patients with a recurrence. This study confirmed previously reported CNVs in either chromosome 12, 14 or 22 in 22/27 patients (81%) [12,13,14,15,16,17,18]. Trisomy 12 is also often detected in other sex cord-stromal tumors and is usually the single copy number alteration in benign sex cord-stromal tumors such as fibromas and thecomas [25]. Monosomy 22 was identified as the sole anomaly in a mixed germ cell sex cord-stromal tumor of the ovary and in a fibrothecoma [26,27], and trisomy 8 as the single copy number variant in a Sertoli-Leydig cell tumor [28]. The effect of concomitant gain of chromosome 14 and loss of 22 in AGCTs is unknown and requires further investigation. In our cohort, copy number alterations are equally present in primary tumors and recurrences. It therefore remains unclear whether chromosome gains and losses are a cause or a consequence of tumor evolution.

### 2.3. Mutational Signatures in AGCTs Are Related to Ageing and Platinum Treatment

In addition to the number of mutations detected per sample, we investigated which mutational processes generated specific single base substitutions (C>A, C>G, C>T, T>A, T>C, and T>G). We applied de novo signature extraction and compared these signatures to COSMIC mutational signatures version 3 (Figure 2a,b). Four different major signatures were identified in the tumor samples. Two of the derived signatures were related to normal ageing processes being present in all cells (COSMIC 3, 5, 37 and COSMIC 3, 5, 40, respectively), one to platinum treatment (COSMIC 31, 35) and one signature with a yet unestablished cause (COSMIC 4, 20, 38, 45). No signatures related to microsatellite instability (MSI) or homologous recombination (HR) deficiency were detected (see Methods). Patients treated with chemotherapy demonstrated a significantly higher total number of variants (median 5463 SNVs) as compared to chemotherapy naïve patients (median 2861 SNVs, *p* < 0.001). Approximately 50% of the variants observed in these chemotherapy-exposed tumors can be explained by the platinum signature. In contrast, two patients (patient 30 and 49) treated with platinum-based chemotherapy did not show single base changes related to platinum treatment. These patients received only three out of six cycles of chemotherapy due to disease progression during treatment, whereas the patients that do show the platinum signature had a prolonged platinum exposure and received up to 2 × 6 cycles of chemotherapy. A recent study established that the contribution of platinum signature is dependent on the duration of treatment [24]. The absence of the platinum signature in these patients could possibly be explained by their resistance to platinum and the short duration of treatment.

### 2.4. Variants in Known Cancer Genes Were Detected in FOXL2, TERT, KMT2D, PIK3CA and TP53

A total of 239 variants in known cancer genes from COSMIC [29] were detected in our AGCT cohort (Appendix A). Most recurrently mutated genes included *FOXL2*, *TERT*, *KMT2D*, *PIK3CA* and *TP53* (Figure 2d). We confirmed the *FOXL2* c.402C>G mutation in 29/33 patients (88%). Two patients had a second mutation in *FOXL2* consisting of a frameshift mutation and a variant in the UTR5 region, respectively (Appendix A). *TERT* C228T and C250T promoter mutations were present in respectively eight and three patients (together 33%), and were mutually exclusive. These variants were present in 2/11 (18%) patients with a primary tumor and in 9/22 (41%) of the patients with a recurrence. The majority of patients with active disease or who died of disease harbored a *TERT* promoter variant (7/12, 58%), while only 3/18 patients with no evidence of disease had a *TERT* promoter mutation (17%, Figure 2d,e). Although the two TERT promoter variants are known hotspots in cancer, previously only the C228T variant had been identified in AGCTs. However, our study and a recently published study have also detected the presence of C250T in AGCTs [30]. Our results corroborate the findings of a previous study that detected the TERT C228T variant more often in recurrences (41%) than in primary tumors (22%) and associated this mutation with impaired prognosis [20].

### 2.5. Subgroup of Patients with High-Grade AGCT Characterized by TP53 Mutation

Out of the five patients with the highest mutational load, three had a *TP53* mutation combined with loss of heterozygosity (R248G, H179R and C135Y, respectively, Figure 2). These tumors harbored numerous copy number alterations, and increased mitotic activity was seen on hematoxylin and eosin (H&E) slides from two patients (18 and 70 per 2 mm^2^, respectively) as compared to *TP53*-wildtype patients (range 2–12 per 2 mm^2^) (Figure 3). The *TP53* mutant tumors harbored a higher number of both SNVs (median 12,027, range 7100–21,452) and SVs (median 188, range 66–314) as compared to the *TP53*-wildtype tumors (median 3336 SNVs, range 1346–9211, median 20 SVs, range 0–120, Figure 2c, Figure 4 and Appendix A). Notably, none of the TP53 mutant samples harbored a TERT promotor mutation, which is detected in 41% of recurrent AGCTs. These mutually exclusive alterations may indicate a different molecular mechanism for disease progression.

We investigated the clinical disease course in the patients harboring a *TP53* mutant tumor. The first patient (patient 46) was diagnosed with AGCT four years prior to study participation and was re-treated with chemotherapy after multiple surgeries and different hormonal and cytostatic treatment regimens (Appendix A). The second patient (patient 2) had a total of four disease recurrences in eight years and is alive with no evidence of disease. The third patient (patient 16) presented with metastases at diagnosis (stage IIB disease), which is unusual since the vast majority of patients are diagnosed with a primary tumor confined to one ovary. This might indicate an early aggressive behavior of the tumor. The patient is being treated for her fifth recurrence and developed breast cancer as a second primary tumor.

In the literature, cases of AGCTs with areas of high-grade malignant morphology in either the primary tumor or recurrence have been reported. One study detected a *TP53* mutation in these high-grade components of 2/4 *FOXL2* mutant AGCTs and stated that *TP53* mutation likely plays a role in the high-grade transformation [31]. A recent case report describes an aggressive AGCT without a *FOXL2* mutation with strongly positive p53 immunohistochemistry and numerous mitoses [32], possibly similar to our finding of one *FOXL2*-wildtype *TP53*-mutant AGCT with high mitotic activity. *TP53* mutations have previously been found in 4–9% of AGCTs [13,14,33]. In conclusion, we found an association between the occurrence of a *TP53* mutation, high mutational burden, copy number alterations and mitotic activity. These characteristics define a subgroup of high-grade AGCTs with a potentially more aggressive tumor behavior.

### 2.6. FOXL2-Wildtype AGCTs Resemble FOXL2-Mutant AGCTs

Four clinically and histopathologically confirmed AGCTs did not harbor the specific *FOXL2* variant (Figure 2d). As described above, we detected a damaging *TP53* variant in one *FOXL2* wildtype tumor. Interestingly, in one other tumor both the *TERT* C228T variant and two *DICER1* variants were found (E1813D and a stop-gain variant Q1230 * predicted to result in nonsense-mediated decay [34], Appendix A). The *TERT* C228T variant has been previously detected in 26% of AGCTs [20] and similarly in 8/33 (24%) patients in our study. *DICER1* variants are detected in 2.38% of all cancers with non-small cell lung cancer, colorectal cancer, endometrial cancer, breast cancer, and melanoma having the greatest prevalence [35]. Somatic heterozygous mutations in *DICER1* are present in 29% of non-epithelial ovarian cancers, including Sertoli-Leydig cell tumors (60%, the majority harboring a E1705K or D1709N hotspot mutation), juvenile GCTs (7%), yolk sac tumors (13%) and mature teratomas (12%) [36]. Specifically, *DICER1* mutations containing nonsense and missense variants in one of the four catalytic residues in the RNase IIIB domain (E1705, D1709, D1810, E1813), as we find, are the major known *DICER1* events in cancer [37]. No germline or somatic exonic truncating or missense *DICER1* variants were detected in the three remaining *FOXL2*-wildtype patients. *FOXL2*-wildtype patients also harbored somatic mutations in the cancer genes *ARID1B, STK11, TP53, PIK3R1, LRP1B, GATA1, NOTCH2, CTNNB1* and *PAX3* (Appendix A). However, none of the *FOXL2*-wildtype patients shared a variant in the same gene. Additionally, no SVs were detected in the vicinity of *FOXL2* (+/−10,000 base pairs) with a predicted effect on expression and/or regulation of *FOXL2*. 

*FOXL2* is thought to be a major tumor driver in GCT, although the absence of a *FOXL2* mutation in a small subset of AGCTs suggests potential alternative mechanisms for AGCT tumorigenesis. In a recent study, GCT tumorigenesis was associated with the combined inactivation of p53 and Rb pathways, with FOXL2 still present in newly developed AGCTs and *FOXL2* downregulation starting during AGCT growth [38]. Cluzet et al. suggest that *FOXL2* downregulation may be a late event in GCTs, and therefore, possibly not a prerequisite for tumor development, but rather for tumor growth. This finding supports the hypothesis of alternative mechanisms for AGCT tumorigenesis.

The copy number profiles, number of SNVs and SVs of the *FOXL2*-wildtype tumors are similar to the tumors with *FOXL2* mutation (Figure 1 and Figure 4). *FOXL2*-wildtype tumors harbored 2791 SNVs (median, range 1919–12,027) and *FOXL2*-mutant 3662 SNVs per tumor (median, range 1346–21,452). Three out of four *FOXL2*-wildtype patients had CNVs in both chromosome 12 and 14 and one patient had additional loss of chromosome 22. One patient had monosomy X, which was previously detected in 1/17 (6%) AGCTs [15] and present in 3/28 (11%) of patients in our cohort. In all *FOXL2*-wildtype tumors, the diagnosis AGCT was reconfirmed by the pathologist. For example, the diagnosis Sertoli-Leydig cell tumor was considered and excluded in the *DICER1* mutated tumor because the tumor harbored the pathological characteristics of an AGCT (Figure 3b) and did not show signs of another ovarian tumor. In addition, this patient was postmenopausal when the ovarian mass was diagnosed, while the peak incidence of a Sertoli-Leydig cell tumor lies around the age of 25, and also they had pre-operatively elevated circulating inhibin and estrogen levels. Our study confirms that the absence of the *FOXL2* c.402C>G mutation does not exclude the diagnosis AGCT. Moreover, in two patients, we identified *TP53* and *DICER1*, respectively, as potential tumor drivers. These findings suggest there may be alternative mechanisms for AGCT development beside the *FOXL2* c.402C>G mutation.

### 2.7. Investigation of Shared Variants between AGCT Patients Confirms a Limited Number of Recurrent Mutations

#### 2.7.1. Overlapping Single Nucleotide Variants

Besides the specific missense variant in *FOXL2*, no overlapping somatic variants between ≥4 AGCT patients were detected in the coding regions of the genome. All variants detected by WGS with a CADD score >5 (Appendix A) were analyzed. Hereby, we identified 39 shared variants between ≥3 patients including the two *TERT* promoter variants (Appendix A). The sole detected coding variant shared by three patients, *TSPYL2* L34Pfs*28, was concluded to be a technical artefact upon Sanger sequencing validation. We identified 305 non-coding variants present in only two patients. However, it remains challenging to interpret the effect of potential non-coding mutations on gene function. Variation in the non-coding areas of the genome is incompletely characterized and may be a result of sequencing and mapping artefacts as well as poorly understood localized hypermutation processes, and the functional effect that these variants may have on cis or trans regulatory elements is unknown [39,40]. Our study confirms the lack of overlap in mutations in AGCTs, as shown in previous studies [13,14]. These previous studies suggest that “second-hit” mutations leading to recurrence may be random.

#### 2.7.2. Overlapping Gene Loss and Structural Variants

Additionally, we investigated if specific gene loss was recurrent across patients. We detected full loss of single or multiple genes in 7/33 (21%) of patients, although there was no overlap in deleted genes between patients. Finally, we investigated overlapping structural variants (other than CNVs) across patients and identified specific breakpoints on chromosome 1 in nine patients (Appendix A). 

### 2.8. Novel Candidate Gene Analysis

Cohort analysis resulted in the identification of 13 genes with mutations in ≥3 patients that could possibly be deleterious, as predicted by the CADD PHRED score >5 (Appendix A). This list included two previously identified genes involved in AGCTs, *FOXL2* and *KMT2D*, and the newly described gene *TP53*. Our study confirms the previously reported limited number of recurrent mutations in individual genes in AGCTs [14].

### 2.9. Intra-Patient Comparisons Reveal Tumor Heterogeneity 

To investigate tumor heterogeneity and tumor evolution over time, tumor tissue from multiple time points and/or multiple tumor locations of five different patients was sequenced and analyzed. In patients 8 and 23, we investigated tumor tissue obtained from two different recurrences and detected that 54–80% of the somatic variants (SNVs and INDELs) in these tumors were unique to a single time point (Appendix A). Surprisingly, two samples taken from the left and right side within a primary AGCT (patient 22) had the least overlap and carried 75–80% unique somatic variants (456 overlapping and 1377–1786 unique variants). The SVs also differed in this early stage of disease (1 and 2 SVs with no overlap, respectively, Appendix A). We investigated the variants in 7 and 5 different recurrences and tumor locations of patient 11 and 13, respectively (Figure 5). These patients showed a similar pattern of distribution between the proportion of overlapping (25–41% and 25–35%) and unique variants (29–73% and 31–66%). The detected larger structural variants also capture the heterogeneity of this tumor, with only 1 and 4 shared SVs between all samples of patient 11 and 13, respectively (Appendix A). Interestingly, all samples from patient 13 share the same SVs present at the initial time point and acquired additional SVs over time.

Patient 11 had active recurrent disease during the study period and required alternative treatment options after multiple surgeries and different chemotherapy and hormone treatment combinations (Figure 6). In this patient, recurrent *PIK3CA* mutations were found and this gene was identified as a potential target for future treatment. However, during the course of her disease, different clonal variants in the *PIK3CA* pathway emerged and disappeared. Reads with the alternative allele, suggesting the presence of a subclonal mutation, were not detected in any of the other samples of this patient at these locations (sequencing depth mean = 32, range 19–43). The PI3K/AKT pathway was thought to be involved in AGCT tumorigenesis since PI3K activity within oocytes irreversibly transforms granulosa cells into AGCTs in mice [41]. Mutations in this pathway, however, have been detected in only a small proportion of patients (2/33, 6% in our study). Although this pathway was recurrently hit in this patient, it remains difficult to assess whether these mutations are drivers or passenger mutations. A previous study identified *KMT2D* inactivation as a driver event in AGCTs and suggest that mutation of this gene may increase the risk of disease recurrence [13]. In our study, mutations in *KMT2D* were present in 6/33 patients (18%). Patient 13 had different exonic inactivating mutations in *KMT2D* in 3/5 tumor samples (Figure 6). In this patient, *KRAS* was the only additional cancer gene besides *FOXL2* and *TERT* that was mutated in all samples. These examples illustrate both inter- and intra-patient heterogeneity in AGCTs.

We also compared the dispersion of mutational signatures within the longitudinally obtained tumor samples from patient 11 and 13. Patient 11 was treated with chemotherapy after the first two tissue samples had been obtained (Figure 6). Only the tissue samples obtained at later recurrences showed the platinum signature (Figure 2b), while tissue obtained before treatment did not show the platinum signature. In line with these results, patient 13 had not been treated with chemotherapy before or during the study and none of her tumor samples harbored the platinum signature. These data confirm that chemotherapy can increase the number of variants, induces specific base changes and potentially plays a role in the heterogeneity between tumor metastases [24]. The lack of overlapping variants within and between patients provides evidence for tumor heterogeneity. This challenges the paradigm of AGCT as being a homogenous tumor. Our study shows that AGCTs, despite the microscopically homogenous cell population, are not homogenous in the genomic alterations they harbor. This can be a challenge for designing effective treatment strategies.

## 3. Discussion

This is the first study to investigate the whole genome of a large cohort of AGCTs by sequencing both tumor and matched reference DNA. AGCT patients did not share specific variants or affected genes except for the *FOXL2, KMT2D* and *TERT* promoter variants. A higher mutational burden was found in recurrent tumors, as compared to primary AGCTs. The molecular differences we detected between and within patients confirm tumor heterogeneity which is an established characteristic of cancer and rejects the view of AGCTs as being a homogenous tumor [42]. The few structural variants detected in AGCTs indicate that DNA repair mechanisms are still intact in these tumors. This study also illustrates the complexity of treating recurrent AGCTs, emerging with decreasing time intervals, and therefore suggests it should not be described as an indolent tumor. We confirm that absence of the *FOXL2* c.402C>G mutation does not exclude AGCT diagnosis and identified *TP53* and *DICER1* as potential drivers in these tumors. Furthermore, we define a subgroup of high-grade AGCTs characterized by a damaging *TP53* mutation, high tumor mutational burden and mitotic activity. 

Whole genome sequencing has emerged as a comprehensive test over the past decade, enabling the detection of both exonic, intronic, and intergenic variation. Recent studies suggest that deeper sequencing (e.g., 90×) of the genome is needed to detect subclonal events with a low allele frequency in tumors and that multi-region sampling of the same tumor is necessary to capture the majority of variation present in a single sample [43,44]. In our study, clonal tumor mutations could reliably be detected by 35× coverage of tumor samples with matched reference DNA. Moreover, the AGCTs in our study were high in tumor content, with the majority of tissues having a tumor percentage of 80–90% (31/46 samples). Variant allele frequency (VAF) calling at 30× in 80–90% tumor samples might be more accurate than VAF calling at 90× in a 30% tumor sample. As a result, this coverage is sufficient to detect clonal mutations in our cohort of high tumor purity samples and is currently used in large cancer studies [45,46]. Comprehensive deep- and multi-region sequencing of tumor samples will be necessary to detect the majority of mutations present in AGCTs and will most likely reduce intra-patient heterogeneity with the detection of all clonal and subclonal events within individual samples. However, the presence of heterogeneity amongst this tumor type, which has previously been defined as homogenous, is an important feature and has implications when searching for novel targets for treatment in these patients. 

The lack of overlapping variants in targetable cancer genes in AGCTs indicates the need for personalized treatment. TERT is essential for the maintenance of telomere length, and thus, influences cellular immortality. TERT activity is an important mechanism for cancer to escape apoptosis and a promising therapeutic target for cancer, as it is highly expressed in most tumor cells and hardly expressed in normal cells. *TERT* mutations have frequently been detected in many cancers [47], and in our study, a damaging promoter variant was consistently present in 33% of patients and in 41% of recurrences. Although it can be difficult to target a promoter variant, TERT silencing has been successful in NRAS mutant melanoma and might be applied to other tumors in the future [48].

In epithelial ovarian cancer, the presence of a *TP53* mutation determines high-grade disease. It can be hypothesized that AGCT patients harboring characteristics of a more aggressive tumor, such as a *TP53* mutation, might respond better to chemotherapy than patients without this variant since chemotherapy targets rapidly dividing cells. This requires further investigation in a larger cohort of *TP53*-mutant and wildtype tumors. 

In conclusion, AGCTs harbor significant genetic heterogeneity between and within patients, and are not a homogenous and stable tumor type. This heterogeneity can be a challenge for future targeted treatment in AGCT patients, and suggests that a personalized, genotype-guided approach is required. Future personalized in vitro drug screens on patient-derived tumor tissue could facilitate the identification of potential patient-specific treatment and targeted sequencing of these tumors could identify actionable mutations.

## 4. Materials and Methods 

### 4.1. Patient Cohort and Study Inclusion Criteria

A national prospective study was performed to obtain patient-derived fresh frozen tumor tissue and corresponding germline DNA (blood or saliva). Patients were included consecutively during their hospital consultation in 5 hospitals between 2018–2019. In addition, patients with fresh frozen tumor material available in the hospital’s pathology archive were asked for consent. Ethical approval was obtained by the Institutional Review Board of the University Medical Center Utrecht (UMCU METC 17-868) and by the board of directors of the participating centers. All participants provided written informed consent. Clinical data were acquired from patient reports.

### 4.2. Tissue Acquisition

All tumor material was obtained directly from surgery, fresh frozen and transported in dry ice and stored in the −80 degrees freezer. From the tissue, 20 × 5 μm slices were cut for DNA isolation. Hematoxylin and eosin (H&E) staining slides were made and reviewed by a pathologist (GNJ) to confirm AGCT diagnosis and assess tumor percentage. Minimal tumor percentage for study inclusion was 40%, with the majority of tissues having a tumor percentage of 80–90% (31/48 samples, Table 1). In parallel, fresh frozen tumor material was crushed in a liquid nitrogen cooled mortar using a pestle, and pulverized tissue was collected for DNA isolation. Material for normal reference DNA isolation was acquired by prospective blood drawn during patient follow-up visits. In cases where patients no longer required follow-up at the hospital, Oragene-DNA OG-500 saliva DNA isolation kits (DNA Genotek, Ottawa, ON, Canada) were mailed to the patient’s residence and taken by the patient after instructions, so that a hospital visit and blood draw were not required.

### 4.3. DNA Isolation, Quantification and Qualification

DNA from the fresh frozen pulverized tumor tissue was isolated using Genomic-tip 100/G (Qiagen, Venlo, NL, USA). Isolated DNA was quantified by Qubit 2.0 dsDNA broad assay kit (Thermo Fisher Scientific, Waltham, MA, USA) and ensured to be of high molecular weight by visualization on a 1.5% agarose gel. Samples were then concentrated if needed to a minimal concentration of 15 ng/μL and a total of 1 μg was aliquoted for WGS. Normal reference DNA isolation from blood was performed according to the DNeasy blood isolation protocol (Qiagen, Venlo, NL, USA) and normal reference DNA isolation from saliva samples performed according to the prepIT-L2P protocol (DNA Genotek, Ottawa, ON, Canada). 

### 4.4. Whole-Genome Sequencing and Variant Calling

DNA was sent for 2 × 150 bp paired-end sequencing on Illumina HiSeq X or NovaSeq 6000 instrument (Illumina, San Diego, CA, USA) to the Hartwig Medical Foundation (HMF, Amsterdam, NL, 23 tumor samples with normal reference DNA) or Novogene (Novogene, Beijing, China, 23 tumor samples with normal reference DNA). Mapping and variant calling from raw fastQ reads was performed using a pipeline (v4.8) from the Hartwig Medical Foundation (HMF) installed locally at the UMCU using GNU Guix (https://github.com/UMCUGenetics/guix-additions; https://github.com/hartwigmedical/pipeline [43]). Sequence reads were mapped with Burrows-Wheeler Alignment v0.75a [49] against human reference genome GRCh37. Realignment of insertions and deletions and base recalibration were performed with the Genome Analysis Toolkit (GATK, v3.8.1 [50]). Somatic single nucleotide variants (SNVs) and small insertions and deletions (INDELS) were called with Strelka (v1.0.14) [51]. All SNVs labelled as “PASS” were included in the analysis. The functional effect of the somatic SNVs and INDELS were predicted with SnpEff (v.4.3) [52]. Somatic structural variants (SVs) were called using GRIDSS (v1.8.0) and copy number variation called by using PURPLE [53].

### 4.5. WGS Data Analysis

The tumor mutational burden (TMB; mutations per Mb) for each sample was derived by dividing the sum of mutations across the entire genome (SNVs, MNVs (multiple nucleotide variants) and INDELS) by the total mappable sequence length of the GRCh37 FASTA file (2858674662) divided by 10⁶, as has been described previously [54]. Wilcoxon rank-sum test was performed for between-group comparisons, two tailed, with a 0.05 significance level.

### 4.6. CNV Analysis

The results obtained from PURPLE (CNVs per breakpoint and per gene) were processed and plotted using in-house R tools for inter- and intra-patient comparison of CNVs. Samples not fulfilling PURPLE quality control criteria for CNV calling (status: “FAIL_SEGMENT”) were not plotted. PURPLE uses allele frequency data (bam reads) for breakpoint detection. In a subsequent quality control step, it checks the result from the SV caller (GRIDSS) to increase evidence for each breakpoint. It tolerates a number of breakpoints without proper SV evidence but flags a sample when that number exceeds a threshold. In samples with potential decreased CNV calling reliability, the SNV calling is unaffected.

### 4.7. Exonic Variant Analysis

The exonic and UTR variant analysis was performed using Alissa (Agilent Technologies Alissa Interpret v5.1.4). Variants in the exonic regions, ±100 base pairs into the intronic regions, and in the 5’ and 3’ UTR of Refseq transcripts were retained for analysis. Variants present in the COSMIC Cancer Gene Census (release v89) as somatic mutations in cancer were annotated (Appendix A). 

To find novel candidate genes in which mutations could contribute to oncogenesis and/or recurrence, we performed a cohort analysis to identify genes with a somatic variant in multiple samples. As this study contained multiple samples from individual patients, this variant list was further filtered to remove any genes in which variation occurred solely within samples of a particular patient. Variants were annotated with the CADD PHRED (Combined Annotation Dependent Depletion) [55] score and genes with no or only one variant >5 were removed.

### 4.8. Whole Genome Variant Analysis

Variation throughout the entire genome was assessed to identify recurrent variations and prioritized by CADD score. Briefly, somatic Variant Call Format (VCF) file of each sample was combined into one complete cohort specific VCF containing somatic variation across all samples. As the effect of non-coding variation is difficult to predict, we annotated all variant positions with the CADD PHRED score, a score that is suitable for assessing the effect of exonic, intronic, regulatory and intergenic variation. To exclude mutations with minimal predicted deleteriousness, we selected the variants with a CADD PHRED score > 5 and identified mutations shared by multiple patients. Candidate variants shared by ≥3 patients were validated by Sanger sequencing, as recurrent unannotated variation can be a result of platform or library preparation technical artifacts [56].

### 4.9. Mutational Signature Analysis

Mutational signatures were created using the R-package mutationalPatterns [57]. We derived mutational signatures from the SNV data and selected the optimal number of signatures by inspecting the non-negative Matrix Factorization (NMF) rank survey as described in the vignette of the MutationalPatterns R-package. In detail, we examined the "rss" (residual sum of squares) plot and the "cophronetic" (cophronetic correlation) plot. We checked for an elbow in the rss plot and aimed for a high cophronetic correlation between model and data. We created four signatures per mutation type and compared them to the signatures from COSMIC mutation signatures version 3 using the cosine-similarity measure. Derived signatures were named according to the proposed etiology of the closest COSMIC signatures.

### 4.10. Tumor Heterogeneity Assessment

SNV heterogeneity was assessed by comparing the number of overlapping SNVs for the patients with multiple tumor samples from different time points and/or tumor locations. SV heterogeneity was investigated using in-house tools based on the R-package StructuralVariantAnnotation (Bioconductor version: Release 3.10).

### 4.11. Homologous Recombination

We investigated Homologous Recombination by the CHORD (https://github.com/UMCUGenetics/CHORD) method, which is a random forest model that predicts homologous recombination deficiency (HRD) using the relative counts of specific somatic mutation contexts. The main contexts used by CHORD are small deletions with flanking microhomology and 1–100 kb structural duplications [58].

### 4.12. Data Availability

WGS Binary Alignment Map (BAM) files are available through controlled access at the European Genome-phenome Archive (EGA), hosted at the EBI and the CRG (https://ega-archive.org), with accession number EGAS00001004249. Requests for data access will be evaluated by the UMCU Department of Genetics Data Access Board (EGAC00001000432) and transferred on completion of a material transfer agreement and authorization by the medical ethical committee of the UMCU to ensure compliance with the Dutch ‘medical research involving human subjects’ act.

## 5. Conclusions

This study shows that AGCTs harbor significant genetic heterogeneity and are not a homogenous and stable tumor type. This heterogeneity can be a challenge for future targeted treatment in AGCT patients, and suggests that a personalized, genotype-guided approach is required. We confirm that absence of the *FOXL2* c.402C>G mutation does not exclude AGCT diagnosis and identified *TP53* and *DICER1* as potential drivers in these tumors. Furthermore, we define a subgroup of high-grade AGCTs characterized by a damaging *TP53* mutation, high tumor mutational burden and mitotic activity. 

## Figures and Tables

**Figure 1 cancers-12-01308-f001:**
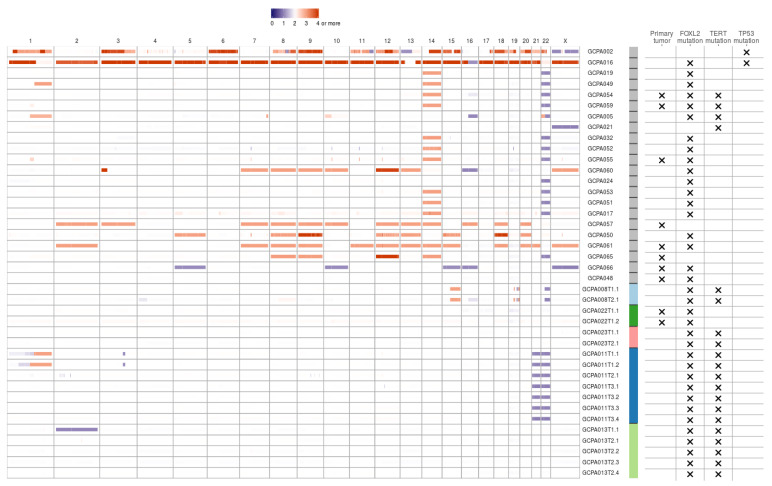
Copy number variants in AGCTs. The majority of copy number alterations are duplications or losses of entire chromosome arms or chromosomes. Blue/purple indicates copy number loss. Orange/red indicates copy number gain. Patients with a TP53 mutation had the most copy number alterations. One sample with a TP53 mutation did not fulfill CNV calling criteria and was therefore excluded from the CNV plot. We identified no differences in CNVs between primary tumors (X) and recurrent samples. Sample IDs: first number indicates time point, second number indicates location (e.g., T3.1 is the first location of the third time point). Sample IDs are ordered by TP53 mutation status, from highest to lowest total number of mutations and patients with multiple samples sequenced are clustered at the end. Color bar: gray indicates one sample per patient, the other colors represent patients with multiple samples.

**Figure 2 cancers-12-01308-f002:**
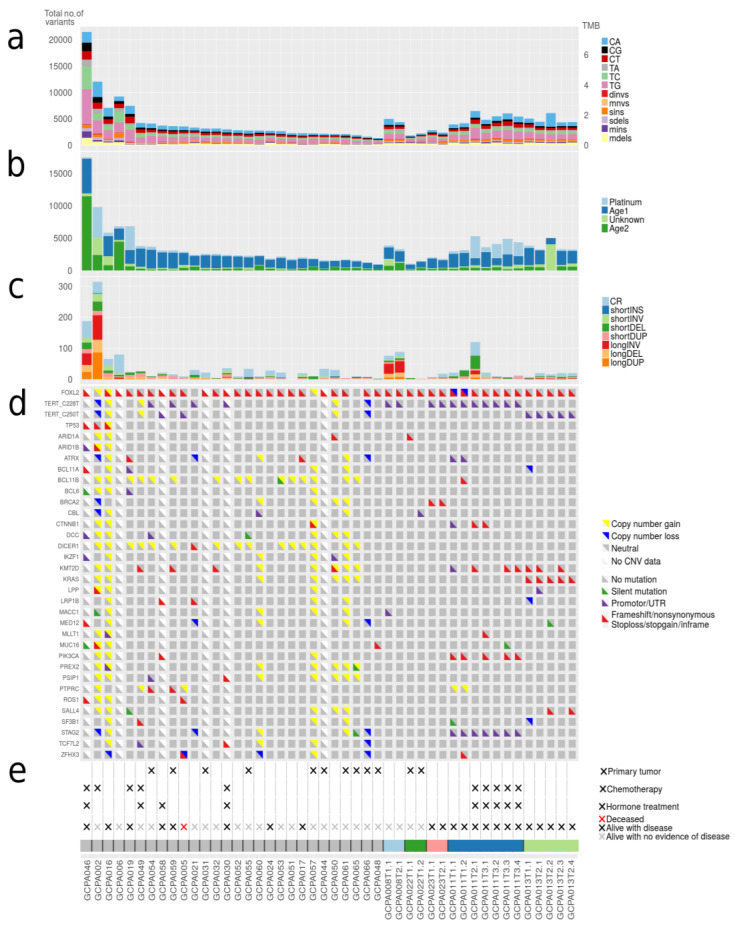
Number of somatic mutations, mutational signatures and variants in cancer genes. (**a**) Total number of small somatic mutations in AGCTs. Dinvs = dinucleotide variants, mnvs = multi-nucleotide variants, sins = single insertions, sdels = single deletions, mins = multiple insertions, mdels = multiple deletions. (**b**) Mutational signatures. AGCTs show mutational signatures related to normal ageing processes (Age1: COSMIC 3, 5, 37 and Age2: COSMIC 3, 5, 40), platinum treatment (COSMIC 31, 35) and one signature with a yet unestablished cause according to COSMIC mutational signatures version 3 (COSMIC 4, 20, 38, 45). (**c**) Total number of structural variants in AGCTs. CR = chromosomal rearrangements, INS = insertions, INV = inversions, DEL = deletions, DUP = duplications. All primary tumors harbor very few structural variants. (**d**) Mutations in genes linked to AGCTs and COSMIC Cancer Genes detected in ≥2 patients are shown. Silent variants include variants +/− 100 bp into intron. (**e**) Clinical parameters and sample IDs. Primary tumors are indicated with (×), the remaining samples are recurrences.

**Figure 3 cancers-12-01308-f003:**
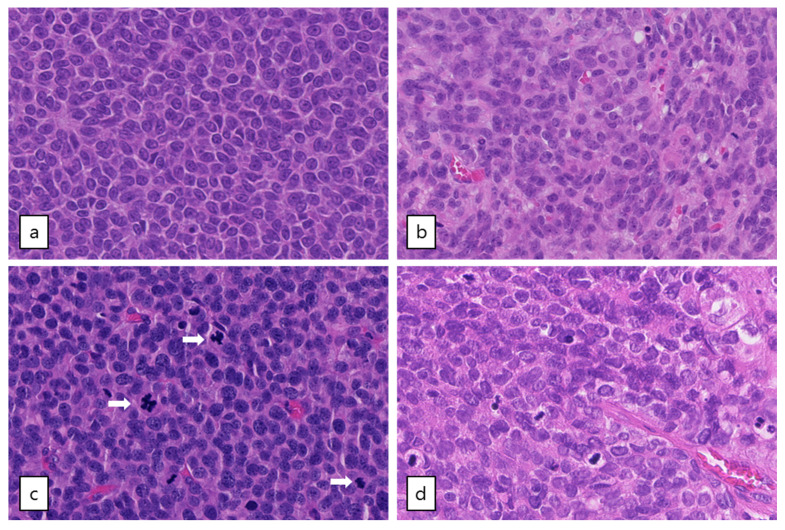
Microscopic examination of AGCTs. H&E staining of adult type granulosa cell tumors showing a homogenous cell population with typical grooved, coffee bean-like nuclei. (**a**) Tumor GCPA011T1.1 with *FOXL2* mutation (20× magnification). (**b**) Tumor GCPA021 (with *TERT* C228T mutation, *DICER1* E1813D and *DICER1* stop-gain variant Q1230 *, 20× magnification). (**c**) and (**d**) Tumors GCPA016 and GCPA002 with pathogenic *TP53* mutation (H179R and C135Y, respectively, 20× magnification). Mitotic activity is indicated by arrows.

**Figure 4 cancers-12-01308-f004:**
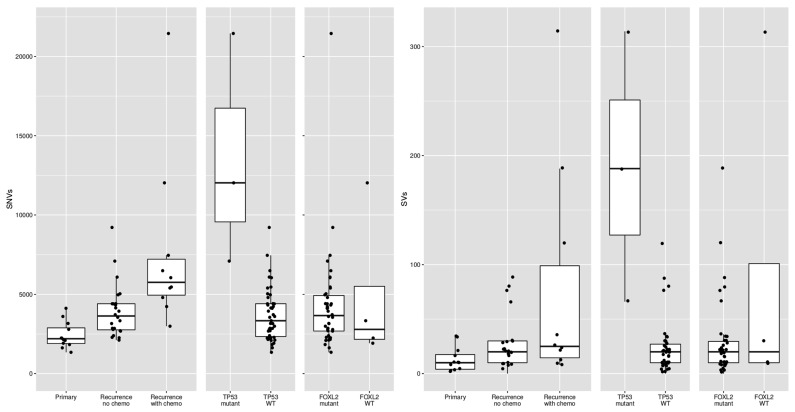
Differences in single-nucleotide variants (SNVs) and structural variants (SVs) between different subsets of AGCTs according to disease phase, prior chemotherapy treatment and mutational status. WT = wildtype.

**Figure 5 cancers-12-01308-f005:**
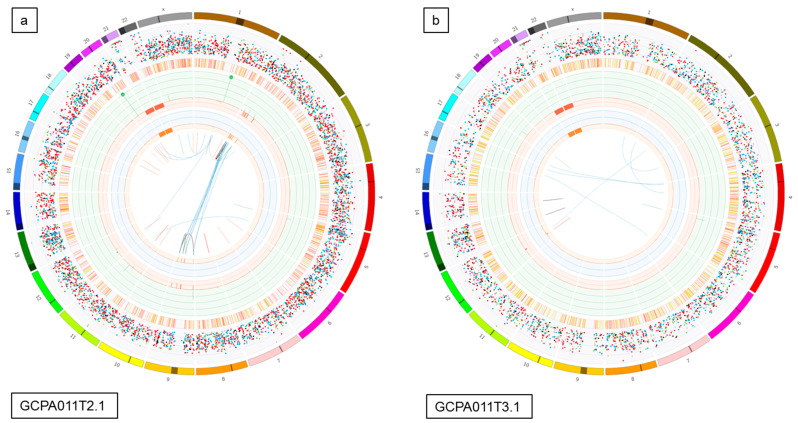
Circos plots and Venn diagrams of intra-patient mutation comparisons. (**a**) and (**b**) Circos plots of samples GCPA011T2.1 and GCPA011T3.1 show differences in structural variants. The outer circle shows the chromosomes, the second circle shows the somatic single nucleotide variants, the third circle shows all observed tumor purity adjusted copy number changes, the fourth circle represents the observed ‘minor allele copy numbers’ across the chromosome, the innermost circle displays the observed structural variants within or between the chromosomes. Translocations are indicated in blue, deletions in red, insertions in yellow, tandem duplications in green and inversions in black. For detailed description of circos plots, see description of the supplementary files. (**c**) and (**d**): Venn diagram from patient 11 and 13 showing the number of unique and overlapping variants between the tumor samples. Intra-patient comparison shows 29–73% unique and 25–41% overlapping variants. The samples GCPA011T2.1 and GCPA011T3.1 have 3117–4809 unique and 1683 overlapping variants.

**Figure 6 cancers-12-01308-f006:**
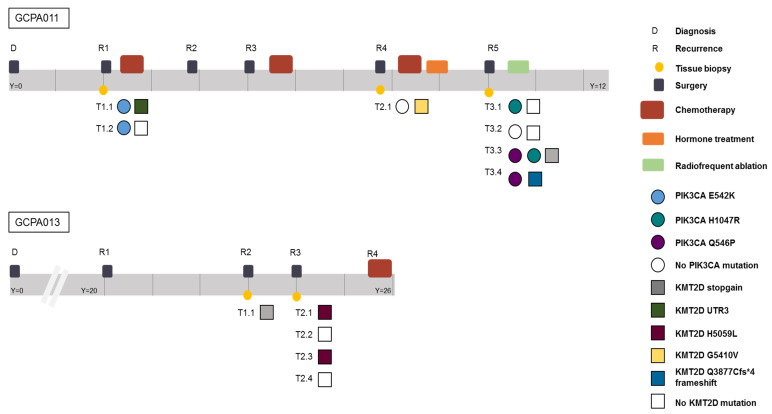
Timeline of patients 11 and 13 capturing the complexity of AGCT treatment and the intra-patient heterogeneity. Different time points and locations harbor different variants in the PIK3CA and KMT2D gene, respectively. Y = years since diagnosis. Both patients are alive with disease.

**Table 1 cancers-12-01308-t001:** Clinical parameters and sample details of adult granulosa cell tumor cohort.

Patient ID	Sample ID	Primary/ Recurrence	Sample Location	Tumor Purity	Reference DNA	Age at Diagnosis	Initial Tumor Stage	Disease Status
GCPA002		Recurrence	abdominal wall	90%	Saliva	35	IC	NED
GCPA005		Recurrence	mesentery	80%	Blood	66	IC	DOD
GCPA006		Recurrence	pelvic wall, left	90%	Blood	37	IC	NED
GCPA008	T1.1	Recurrence	small pelvis	90%	Blood	53	IA	NED
	T2.1	Recurrence	lung	90%	Blood			
GCPA011	T1.1	Recurrence	diaphragm right side	90%	Blood	53	IC	AWD
	T1.2	Recurrence	diaphragm right side	90%	Blood			
	T2.1	Recurrence	small bowel meso	70%	Blood			
	T3.1	Recurrence	liver	80%	Blood			
	T3.2	Recurrence	pelvis right side	60%	Blood			
	T3.3	Recurrence	ligamentum triangulare left	70%	Blood			
	T3.4	Recurrence	greater curvature stomach	60%	Blood			
GCPA013	T1.1	Recurrence	paracolic right	90%	Blood	50	IC	AWD
	T2.1	Recurrence	bladder peritoneum	80%	Blood			
	T2.2	Recurrence	iliac left	60%	Blood			
	T2.3	Recurrence	promontory	70%	Blood			
	T2.4	Recurrence	iliac right	90%	Blood			
GCPA016		Recurrence	abdominal wall	60%	Blood	69	IIB	AWD
GCPA017		Recurrence	below liver	80%	Blood	36	IC	AWD
GCPA019		Recurrence	lateral of psoas muscle	80%	Blood	57	IA	AWD
GCPA021		Recurrence	bladder peritoneum	70%	Blood	61	IC	NED
GCPA022	T1.1	Primary	ovary (left side within tumor)	70%	Blood	57	IA	NED
	T1.2	Primary	ovary (right side within tumor)	80%	Blood			
GCPA023	T1.1	Recurrence	peritoneal cavity	80%	Blood	48	IA-C *	AWD
	T2.1	Recurrence	mesentery rectosigmoid	80%	Blood			
GCPA024		Recurrence	rectosigmoid	60%	Blood	30	IA-C *	AWD
GCPA030		Recurrence	bladder peritoneum	90%	Blood	43	IA-C *	AWD
GCPA031		Primary	ovary left	80%	Blood	49	IA	NED
GCPA032		Recurrence	obturatorius loge right	40%	Blood	61	unknown	NED
GCPA044		Primary	ovary	50%	Blood	61	IC	NED
GCPA046		Recurrence	spleen	90%	Blood	54	IA-C *	AWD
GCPA048		Primary	left ovary	80%	Blood	35	IC	NED
GCPA049		Recurrence	omentum	90%	Blood	52	unknown	NED
GCPA050		Recurrence	pouch of Douglas	60%	Saliva	32	IA	NED
GCPA051		Recurrence	pouch of Douglas	90%	Blood	39	IA-C *	NED
GCPA052		Recurrence	suprarenal infrahepatic	90%	Blood	37	IC	NED
GCPA053		Recurrence	ileocecal	80%	Saliva	39	IA	NED
GCPA054		Primary	ovary	90%	Blood	75	IC	NED
GCPA055		Primary	ovary	40%	Blood	65	IA	NED
GCPA057		Primary	ovary	80%	Saliva	65	unknown	NED
GCPA058		Recurrence	liver	80%	Saliva	53	IA	AWD
GCPA059		Primary	ovary	90%	Saliva	66	IA	AWD
GCPA060		Recurrence	ileum	90%	Saliva	47	IA	NED
GCPA061		Primary	ovary	80%	Saliva	71	IA	NED
GCPA065		Primary	ovary	90%	Blood	61	IA	NED
GCPA066		Primary	ovary	70%	Saliva	29	IA	NED

* Tumor limited to one ovary, no information available on potential capsule rupture before or during surgery. Sample IDs: first number indicates time point, second number indicates location (e.g., T3.1 is the first location of the third time point). NED: No evidence of disease. DOD: Dead of disease. AWD: alive with disease.

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
