# Peer review of "Whole Genome Analysis of Ovarian Granulosa Cell Tumors Reveals Tumor Heterogeneity and a High-Grade TP53-Specific Subgroup"

_cancers, 2020, doi:10.3390/cancers12051308_

Round 1

Reviewer 1 Report

Whole genome analysis of ovarian granulosa cell tumours reveals tumour heterogeneity and a high-grade TP53-specific subgroup

This represents the largest cohort of well characterised adult granulosa cell tumours (AGCT) using medium depth whole genome sequencing (WGS) from high-quality fresh frozen tissue which is, importantly, accompanied by WGS from matched normal samples. The authors aim: to comprehensively characterise the mutational landscape of AGCT; identify subgroups of patients (inter-patient heterogeneity) and identify heterogeneity between tumour samples from the same patient (intra-patient heterogeneity). The cohort is made up of primary tumours from 11 patients (1 of which has two primary tumour sites) and samples from tumour recurrences from a further 22 patients, 4 of which have samples taken from multiple sites at two or more time points. This design allows investigation of intra-tumour heterogeneity using multiple samples from the same patient which is desirable and often not the case even in the small number of patients shown here.

  1. It is notable (as the authors do mention in the discussion) that the average depth of sequencing (30X) here is suboptimal for tumour samples relative to current standard in cancer genomics. It is not my opinion that that should detract from the insights that can be made at this depth, however caveats should clearly be stated where it may affect the results. For example, in the detection of sub-clonal variants which have particular importance in the study of intra-tumour heterogeneity, and in the detection of structural variants, particular inversions, translocations and smaller copy number changes.

  1. The authors aim to comprehensively characterise the mutational landscape of these tumours but they do not specifically describe the patterns of structural variants (not copy number variants (CNVs)) that they observe, with the above limitation accepted. The methods suggest that they used GRIDSS to detect them but the results are not specifically discussed in the text. WGS is a unrivalled resource with which to ask such questions so they should not go unmentioned.

In general, throughout the manuscript the authors refer to SVs but it is not clear whether they mean CNVs, the undiscussed harder to detect SVs or both.

  1. The authors put a lot of emphasis (including the title) on the discovery of a novel TP53 mutant subgroup in AGCT. However, although the results look promising and could be of tremendous clinical impact, it’s worth bearing in mind that the sample size is at most a recurrent sample from each of 3 patients. Also, one of these samples didn’t meet QC criteria to enable copy number variant analysis. It would be good to know more about how this failure could affect other analyses? In general, given the very small N in this subgroup the focus should be on the need for future follow up in larger sample sizes and the immediate conclusions should be toned down.

  1. Throughout the manuscript, samples from recurrent samples are analysed alongside samples from primary tumours. The distinction between these two sets of samples is important and needs clarified throughout. This is particularly true for comparisons between patients as there will be differences between primary tumours and recurrences that confound the between patient differences. For example on line 153, patients treated with chemotherapy had a higher total number variants than treatment naïve patients. However, is this because the chemo treated patients all were recurrences?

  1. In line 287, the authors make the assertion that ‘‘second-hit’ mutations leading to recurrence may be random’. This comes from the observation that particular genes that are fully deleted are not fully deleted in multiple samples. However, firstly is there evidence that these deletions are occurring as second hits? Secondly, is there evidence that they are truly occurring at random or are they more likely to occur at these genes than others? Thirdly, was the main point not that these deletions were not recurrent?

  1. When considering intra-tumour heterogeneity the authors state that the majority of SNVs are unique to a single time point. However, the possibility that these are present as sub-clonal mutations and therefore not necessarily detectable in different samples from the same tumour should be discussed. Particularly true for the example of variants in the PIK3CA pathway emerging and disappearing? Also, using variant allele frequency to support whether these variants are likely to be early/late, clonal/subclonal would lend support to this section.

  1. A more minor subjective comment is that the results section would benefit from more descriptive paragraph headings. That is, each paragraph details the main finding of that paragraph rather than the type of analysis described within.

Author Response

Whole genome analysis of ovarian granulosa cell tumours reveals tumour heterogeneity and a high-grade TP53-specific subgroup

This represents the largest cohort of well characterised adult granulosa cell tumours (AGCT) using medium depth whole genome sequencing (WGS) from high-quality fresh frozen tissue which is, importantly, accompanied by WGS from matched normal samples. The authors aim: to comprehensively characterise the mutational landscape of AGCT; identify subgroups of patients (inter-patient heterogeneity) and identify heterogeneity between tumour samples from the same patient (intra-patient heterogeneity). The cohort is made up of primary tumours from 11 patients (1 of which has two primary tumour sites) and samples from tumour recurrences from a further 22 patients, 4 of which have samples taken from multiple sites at two or more time points. This design allows investigation of intra-tumour heterogeneity using multiple samples from the same patient which is desirable and often not the case even in the small number of patients shown here.

  1. It is notable (as the authors do mention in the discussion) that the average depth of sequencing (30X) here is suboptimal for tumour samples relative to current standard in cancer genomics. It is not my opinion that that should detract from the insights that can be made at this depth, however caveats should clearly be stated where it may affect the results. For example, in the detection of sub-clonal variants which have particular importance in the study of intra-tumour heterogeneity, and in the detection of structural variants, particular inversions, translocations and smaller copy number changes.

We would like to thank the reviewer for his/her time to review this manuscript.

We agree with the reviewer that 30X coverage is unable to reliably detect subclonal events. However, in contrast to many other studies in other cancer types, our samples yield a high tumor percentage, with the majority of tissues having a tumor percentage of 80-90% as determined by the pathologist. Therefore, dilution by normal non-tumor DNA is small – enabling us to detect clonal events at a lower sequencing depth than required for other tumor type sequencing studies with lower tumor purity. However, we do agree that deeper sequencing is always better whenever possible. Future studies at higher depth should further clarify the complexity of the observed clonality.

Tumor heterogeneity is a well-established feature of cancer, although granulosa cell tumors have repeatedly been described as homogenous tumors. We show that between- and within patient heterogeneity also exists in AGCTs. This is an important feature and should be kept in mind when searching for novel targets for treatment in these patients.

To address your and Reviewer 2’s concerns on this topic, we have added the following textual changes in the Discussion:

Line 383

“Recent studies suggest that deeper sequencing (e.g. 90X) of the genome is needed to detect subclonal events with a low allele frequency in tumors and that multi-region sampling of the same tumor is necessary to capture the majority of variation present in a single sample.”

Line 392

“Comprehensive deep- and multi-region sequencing of tumor samples will be necessary to detect the majority of mutations present in aGCTs and will most likely reduce intra-patient heterogeneity with the detection of all clonal and subclonal events within individual samples. However, the presence of heterogeneity amongst this tumor type that has previously been defined as homogenous is an important feature and has implications when searching for novel targets for treatment in these patients.” 

2. The authors aim to comprehensively characterise the mutational landscape of these tumours but they do not specifically describe the patterns of structural variants (not copy number variants (CNVs)) that they observe, with the above limitation accepted. The methods suggest that they used GRIDSS to detect them but the results are not specifically discussed in the text. WGS is a unrivalled resource with which to ask such questions so they should not go unmentioned.

We agree with the reviewer that WGS allows for SV detection and that this is an important aspect of the full molecular characterization of this tumor type. We therefore added the different groups of SVs in Figure 2.

AGCTs harbor very few structural variants (except for the TP53 mutant samples and a few other recurrence samples). The low amount of structural variants detected in AGCTs indicate that DNA repair mechanisms are still intact in these tumors. We added this sentence in line 96 and line 381, respectively.

Also, Supplementary Figure 1 shows full gene loss (i.e. a copy number of less than 0.5) and Figure 2a and b SVs in > 1 sample and SVs with overlapping breakpoints in chromosome 1. To clarify this, we made a subparagraph 2.7.2 on shared structural variants between AGCT patients:

Line 296

“2.7.2. Overlapping gene loss and structural variants

Additionally, we investigated if specific gene loss was recurrent across patients. We detected full loss of single or multiple genes in 7/33 (21%) of patients, although there was no overlap in deleted genes between patients. Finally, we investigated overlapping structural variants (other than CNVs) across patients and identified specific breakpoints on chromosome 1 in nine patients (Supplementary Figure 1, 2a-b).”

In general, throughout the manuscript the authors refer to SVs but it is not clear whether they mean CNVs, the undiscussed harder to detect SVs or both.

We regret that this was not clear. According to the reviewer’s suggestions, we aimed to clarify this in the text with the addition below and by specifically only addressing CNVs in section 2.2.

Line 95

“We identified 20 structural variants (SVs; excluding copy number variants (CNVs)) per sample (median, range 0-314).”

3. The authors put a lot of emphasis (including the title) on the discovery of a novel TP53 mutant subgroup in AGCT. However, although the results look promising and could be of tremendous clinical impact, it’s worth bearing in mind that the sample size is at most a recurrent sample from each of 3 patients. Also, one of these samples didn’t meet QC criteria to enable copy number variant analysis. It would be good to know more about how this failure could affect other analyses? In general, given the very small N in this subgroup the focus should be on the need for future follow up in larger sample sizes and the immediate conclusions should be toned down.

We agree that the number of patients with a TP53 is low in our study, but this is not unexpected. Despite the limited number of publications on mutations in AGCTs, mutations in TP53 have been consistently found in approximately 4-9% of samples. Based on this, in our study with 33 patients, we could anticipate 1-3 patients. One conference abstract (ref 32) reports on 105 AGCTs and found a TP53 mutation in 9% of samples. Also, one study (ref 30) detected a TP53 mutation in high-grade components of 2/4 FOXL2 mutant AGCTs and therefore hypothesized that TP53 mutation likely plays a role in the high-grade transformation. A recent case report (ref 31) describes an aggressive AGCT without a FOXL2 mutation with strongly positive p53 immunohistochemistry and numerous mitoses. These findings strengthen our findings that TP53 mutant AGCTs have characteristics of a high grade tumor.

Although the absolute numbers of samples with TP53 mutations in individual studies on this rare tumor type will remain low, including our study, this could be an important subgroup with a potentially more aggressive tumor behavior that could benefit from different treatment strategies than other AGCTs. We feel therefore, that this is important to emphasize in our manuscript.

One of the samples with TP53 mutation (sample GCPA046) indeed did not meet quality criteria for the CNV caller. The CNV caller that we use (PURPLE) uses allele frequency data (bam reads) for breakpoint detection. In a subsequent QC step, it checks the result from the SV caller (gridSS) to increase evidence for each breakpoint. It tolerates a number of breakpoints without proper SV evidence but flags a sample as "failed QC" when that number exceeds a threshold. The sample GCPA046 has an average number of small SVs but many large chromosomal rearrangements (see circos plot), which is likely causing problems for the QC step in PURPLE. Despite decreased CNV calling reliability in that sample, the SNV calling is unaffected.

To clarify this, we expanded the paragraph on the methods of the CNV calling:

“4.6. CNV analysis

The results obtained from PURPLE (CNVs per breakpoint and per gene) were processed and plotted using in-house R tools for inter- and intra-patient comparison of CNVs. Samples not fulfilling PURPLE quality control criteria for CNV calling (status: “FAIL_SEGMENT”) were not plotted. PURPLE uses allele frequency data (bam reads) for breakpoint detection. In a subsequent quality control step, it checks the result from the SV caller (GRIDSS) to increase evidence for each breakpoint. It tolerates a number of breakpoints without proper SV evidence but flags a sample when that number exceeds a threshold. In samples with potential decreased CNV calling reliability, the SNV calling is unaffected.”

Since we focus on the variants in the TP53 mutant samples, we added the circos plots of these samples to Supplementary Figure 3.

 4. Throughout the manuscript, samples from recurrent samples are analysed alongside samples from primary tumours. The distinction between these two sets of samples is important and needs clarified throughout. This is particularly true for comparisons between patients as there will be differences between primary tumours and recurrences that confound the between patient differences. For example on line 153, patients treated with chemotherapy had a higher total number variants than treatment naïve patients. However, is this because the chemo treated patients all were recurrences?

We thank the reviewer for bringing up this important point. It is indeed interesting to investigate further whether the increased number of variants can be explained by chemotherapy or from the fact that it is a recurrence. In patient 11, all samples used are recurrent disease. The chemo naïve recurrent tissue (GCPA011T1.1 and T1.2) did not show the platinum signature whereas after having received chemotherapy, all samples do show the signature and have an increased number of mutations. We therefore conclude that chemotherapy can increase the number of variants (line 350, as has been shown in other tumors).

To investigate this further, we made a new barplot for the number of SNVs and SVs divided into primary tumor– recurrence without chemotherapy – recurrence with chemotherapy (please find attached Figure 4). Although the recurrence samples treated with chemotherapy have the highest median number of SNVs, there is a significant difference in SNVs between the primary samples and recurrence samples that were not treated with chemotherapy (median 2199.5 vs 3634.5, t-test p-value = 0.0009). In addition, the difference in SNVs in recurrences with versus without chemotherapy was not significant (t-test p-value 0.06). These results suggest that recurrences harbor more variants than primary tumors, also when we stratify for chemotherapy treatment. We added this information to the manuscript line 107 – 112:

“It is known that platinum based chemotherapy can increase the number of variants. However, when we compared the mutations in primary tumors with recurrences that have not been treated with chemotherapy, the number of variants was still significantly different (median 2199 SNVs versus 3634 SNVs, t-test p-value = 0.0009). These results suggest that recurrences, also when stratified for prior chemotherapy treatment, harbor significantly more variants than primary tumors.”

5. In line 287, the authors make the assertion that ‘‘second-hit’ mutations leading to recurrence may be random’. This comes from the observation that particular genes that are fully deleted are not fully deleted in multiple samples. However, firstly is there evidence that these deletions are occurring as second hits? Secondly, is there evidence that they are truly occurring at random or are they more likely to occur at these genes than others? Thirdly, was the main point not that these deletions were not recurrent?

The ‘second-hit’ mutations refer to single nucleotide variants, that seem random since they are not shared between patients. On your suggestion we have now clarified this paragraph in section 2.7.1, now titled “Overlapping single nucleotide variants”.

Line 297

“Our study confirms the lack of overlap in mutations in AGCTs, as shown in previous studies [13,14]. These previous studies suggest that “second-hit” mutations leading to recurrence may be random.”

6. When considering intra-tumour heterogeneity the authors state that the majority of SNVs are unique to a single time point. However, the possibility that these are present as sub-clonal mutations and therefore not necessarily detectable in different samples from the same tumour should be discussed. Particularly true for the example of variants in the PIK3CA pathway emerging and disappearing? Also, using variant allele frequency to support whether these variants are likely to be early/late, clonal/subclonal would lend support to this section.

We agree with the reviewer that this is an important point. We visually checked in IGV whether the three variants identified in the PIK3CA gene were present in the other samples, although with a lower variant allele frequency, suggesting a subclonal mutation. Screenshots from IGV for the locations of the three PIK3CA variants in all samples are attached. We found that the H1047R variant was present in GCPA011T3.3 instead of GCPA011T3.4 (in agreement with Supplementary File 2), so we changed this in Figure 6. Sample GCPA011T3.4 had yet another PIK3CA mutation, so Figure 2d remains unchanged.

We confirmed the presence of the different PIK3CA variants as shown in Figure 6. In addition, in the other samples and/or on the other positions, we did not detect (any) reads for the alternative allele despite good coverage (mean 32, range 19 – 43X). At a minimum coverage of 19X, and with no alternate allele detected, the PIK3CA variant, if present, would have to be present at a variant allele percentage of <5% (assuming 100% tumor purity). To clarify this in the text, we have added the following sentence:

Line 331

“However, during the course of her disease, different clonal variants in the PIK3CA pathway emerged and disappeared. Reads with the alternative allele, suggesting the presence of a subclonal mutation, were not detected in any of the other samples of this patient at these locations (sequencing depth mean = 32, range 19-43).”

7. A more minor subjective comment is that the results section would benefit from more descriptive paragraph headings. That is, each paragraph details the main finding of that paragraph rather than the type of analysis described within.

Great suggestion. We aimed to summarize the main finding of the paragraph as paragraph heading:

 2.1. Description of WGS cohort

2.2. Common copy number alterations in chromosome 12, 14 and 22

2.3. Mutational signatures in AGCTs are related to ageing and platinum treatment

2.4. Variants in known cancer genes were detected in FOXL2, TERT, KMT2D, PIK3CA and TP53

2.5. Subgroup of patients with high-grade AGCT characterized by TP53 mutation

2.6. FOXL2-wildtype AGCTs resemble FOXL2-mutant AGCTs

2.7. Investigation of shared variants between AGCT patients confirms a limited number of recurrent mutations

2.7.1. Overlapping single nucleotide variants

2.7.2. Overlapping gene loss and structural variants

2.8. Novel candidate gene analysis

2.9. Intra-patient comparisons reveal tumor heterogeneity

Reviewer 2 Report

This study analyzed whole genome sequencing data of 46 tumor samples from 33 patients. The authors were able to confirm the findings of published studies on AGCT. While the novelty was thus reduced to some extent, they provided sufficient results to demonstrate tumor heterogeneity of AGCT, which emphasized the importance of personalized treatment of AGCT. Overall this is a good study. Below are a few points that should be addressed.

  1. Reference paper [13] and [14] reported conflicting results regarding TMB difference between primary and recurrent. While both papers used whole-exome data to calculate TMB, it’s unclear if the patients of the two papers received same treatment. Therapies, for example chemo-therapy and radiation-therapy, are known to be able to result in additional mutations. The opposite findings of the two papers may be due to different choice of treatments. And [13] used all SNVs and INDELS, while [14] included only functional mutations. I’d suggest that you group the samples into 3 categories, ie. “primary”, “recurrent after chemotherapy” and “recurrent after surgery”, and compare TMB among the 3 categories using total mutations and then functional mutations.
  2. From both Figure 1 and Figure 2C, it looks like there are some consistent or recurrent CNVs exclusive to “single-sample patients”, for example amplifications in chromosome 7,8,9,12,13,14 shown in Figure 1 and those yellow cells in Figure 2C. It’s possible that such discrepancies may be technical noises which were not ruled out by the pipeline you used. Usually building a pooled reference from normal samples and jointly calling CNVs for tumor samples would generate CNVs with higher confidence and accuracy. So please provide detailed description of the methods and steps you used, and confirm if the pattern I described is due to technical confounders or may reflect real biological signal. And please provide some explanation or discussion about the “biological signal” if you think that signal truly exists. 
  3. And it has been proven that single tumor sample is not sufficient to capture full mutation spectrum of tumor tissues. Usually the number of unique mutations called from WGS/WES for one patient may start to saturate when 5 or more tumor samples are used. So it’s highly likely that you may be able to observe a larger number of high frequency mutations among patients (not samples). And the patient level heterogeneity are likely less significant than what you’ve observed between samples. (You can find some useful information from https://www.ncbi.nlm.nih.gov/pmc/articles/PMC6890490/ method section “Multiregion whole-exome sequencing analysis”). So it will be good if you could discuss the utility of multi-region or multi-time-point samples and how it would help confirm or supplement your original findings in your discussion section.
  4. In line 175, you said this study is the first to describe the C250T variant in AGCTs. But at least one paper reported similar findings (https://www.nature.com/articles/s41379-020-0514-3). You should cite this paper or earlier ones.

Author Response

1. Reference paper [13] and [14] reported conflicting results regarding TMB difference between primary and recurrent. While both papers used whole-exome data to calculate TMB, it’s unclear if the patients of the two papers received same treatment. Therapies, for example chemo-therapy and radiation-therapy, are known to be able to result in additional mutations. The opposite findings of the two papers may be due to different choice of treatments. And [13] used all SNVs and INDELS, while [14] included only functional mutations. I’d suggest that you group the samples into 3 categories, ie. “primary”, “recurrent after chemotherapy” and “recurrent after surgery”, and compare TMB among the 3 categories using total mutations and then functional mutations.

We would like to thank the reviewer for his/her time to review this manuscript. In the introduction, we compared the total amount of variants found in the studies. We agree with the reviewer that therapy can increase the number of mutations found in a tumor sample. Therefore, we have now, on your suggestion changed Figure 4 and divided the barplots for SNVs and SVs into primary samples  – recurrence without chemotherapy – recurrence with chemotherapy samples, according to the reviewer’s suggestions. We compared the SNVs between the groups. Although the recurrence samples from patients treated with chemotherapy have the highest median number of SNVs, there is a significant difference in SNVs between the primary samples and recurrence samples that were not treated with chemotherapy (median 2199.5 vs 3634.5, t-test p-value = 0.0009). The difference between SNVs in recurrences in patients treated with or without chemotherapy was not significant (t-test p-value 0.06), but the subgroups are small. These results show that recurrences harbor more variants than primary tumors, and increases even more after chemotherapy. We added this to the article at line 107 – 112: “It is known that platinum-based chemotherapy can increase the number of variants. However, when we compared the mutations in primary tumors with recurrences that have not been treated with chemotherapy, the number of variants was still significantly different (median 2199 SNVs versus 3634 SNVs, t-test p-value = 0.0009). These results suggest that recurrences, also when stratified for prior chemotherapy treatment, harbor significantly more variants than primary tumors”.

2. From both Figure 1 and Figure 2C, it looks like there are some consistent or recurrent CNVs exclusive to “single-sample patients”, for example amplifications in chromosome 7,8,9,12,13,14 shown in Figure 1 and those yellow cells in Figure 2C. It’s possible that such discrepancies may be technical noises which were not ruled out by the pipeline you used. Usually building a pooled reference from normal samples and jointly calling CNVs for tumor samples would generate CNVs with higher confidence and accuracy. So please provide detailed description of the methods and steps you used, and confirm if the pattern I described is due to technical confounders or may reflect real biological signal. And please provide some explanation or discussion about the “biological signal” if you think that signal truly exists. 

We agree with the reviewer that it is important to rule out the possibility of technical noise. Therefore, we plotted CNVs of high grade serous ovarian cancer samples (not included in our study) together with AGCTs (included in our study). All samples were processed using the same software. This plot, shown below, does not suspect any technical noise since the high grade ovarian cancers have gains and losses throughout the genome. It strengthens the evidence that the amplifications shown in Figure 1 are specific for the AGCT samples.

We considered the effect of the chromosome gains and losses on genes. For example,

gains in chr 14 could lead to upregulation of AKT1 and BCL11B and loss of chr 22 in downregulation of CHEK2. However, we feel there is not enough known on the direct result of a chromosome gain or loss on gene expression and/or protein level. We therefore did not speculate on this point in the manuscript.

For completeness, we now added the chromosomes gained in at least 15% of samples to the text:

Line 119

“and gain of chromosome 1, 7, 8, 9, 12, 13, 14, 15, 18 or 20 were identified in at least 15% of patients.”.

3. And it has been proven that single tumor sample is not sufficient to capture full mutation spectrum of tumor tissues. Usually the number of unique mutations called from WGS/WES for one patient may start to saturate when 5 or more tumor samples are used. So it’s highly likely that you may be able to observe a larger number of high frequency mutations among patients (not samples). And the patient level heterogeneity are likely less significant than what you’ve observed between samples. (You can find some useful information from https://www.ncbi.nlm.nih.gov/pmc/articles/PMC6890490/ method section “Multiregion whole-exome sequencing analysis”). So it will be good if you could discuss the utility of multi-region or multi-time-point samples and how it would help confirm or supplement your original findings in your discussion section.

Thank you for this information. In this study, we want to emphasize heterogeneity in AGCTs.

Although tumor heterogeneity is a well-established feature of cancer, granulosa cell tumors have repeatedly been described as homogenous tumors. We show that between- and within patient heterogeneity also exists in AGCTs. This is an important feature and should be kept in mind when searching for novel therapeutic targets in these patients. We agree that a comprehensive deep coverage and multi-region sequencing approach is needed to fully characterize the presence of heterogeneity within and amongst patients, although this is beyond the scope of this study. We have now modified the text in two locations in the discussion to address this point and also emphasize that the presence of heterogeneity within this cancer type will have implications for patient treatment. We have also added the suggested article as a reference.

Textual modifications:

Line 383

“Recent studies suggest that deeper sequencing (e.g. 90X) of the genome is needed to detect subclonal events with a low allele frequency in tumors and that multi-region sampling of the same tumor is necessary to capture the majority of variation present in a single sample.”

Line 392

“Comprehensive deep- and multi-region sequencing of tumor samples will be necessary to detect the majority of mutations present in aGCTs and will most likely reduce intra-patient heterogeneity with the detection of all clonal and subclonal events within individual samples. However, the presence of heterogeneity amongst this tumor type, that has previously been defined as homogenous, is an important feature and has implications when searching for novel targets for treatment in these patients.

4. In line 175, you said this study is the first to describe the C250T variant in AGCTs. But at least one paper reported similar findings (https://www.nature.com/articles/s41379-020-0514-3). You should cite this paper or earlier ones.

We changed the text according to the reviewer’s suggestions and cited this paper in the text.

Line 179

 “Previously only the C228T variant had been identified in AGCTs, however, our study and a recently published study have also detected the presence of C250T in AGCTs.”

Round 2

Reviewer 2 Report

The authors addressed both major and minor points.